# EMPATIA: A Guide for Communicating the Diagnosis of Neuromuscular Diseases

**DOI:** 10.3390/ijerph19169792

**Published:** 2022-08-09

**Authors:** Isabella Araujo Mota Fernandes, Renata Oliveira Almeida Menezes, Guilhermina Rego

**Affiliations:** 1Faculty of Medicine, University of Porto, 4099-002 Porto, Portugal; 2Lauro Wanderley University Hospital, Federal University of Paraíba, Joao Pessoa 58051-900, Brazil; 3Law Department, Federal University of Rio Grande do Norte, Caico 59300-000, Brazil

**Keywords:** truth disclosure, neuromuscular diseases, physician-patient relations, communication

## Abstract

Introduction: Neuromuscular diseases comprise a heterogeneous group of genetic syndromes that lead to progressive muscle weakness, resulting in functional limitation. There is a gap in the literature regarding the communication of the diagnosis of such diseases, compromising the autonomy of patients and families, besides causing stress on the assistant physician. Objectives: Developing a guide to reduce communication barriers in the diagnosis of neuromuscular diseases. Methodology: Systematic review, after searching the descriptors (“Muscular Diseases” OR “Neuromuscular Diseases”) AND (“Truth Disclosure” OR “Bad news communication” OR “Breaking bad News”) in the Pubmed, Bireme, and Scopus websites, and these results were analyzed through narrative textual synthesis. Results: 16 articles were submitted to the final analysis, giving rise to seven steps to support the communication process. These are Empathy, Message, Prognosis, Reception, Time, Individualization, and Autonomy. Discussion and conclusion: The empathic transmission of the message and the prognosis must accommodate the feelings of the interlocutors with different information needs. In this way, communication planning optimizes the time and individualizes each context, respecting the autonomy of those involved. EMPATIA reflects the bioethical and interdisciplinary analysis of the literature and comes to fill the gap related to the communication of bad news in neuromuscular diseases.

## 1. Introduction

Neuromuscular diseases comprise a heterogeneous group of genetic syndromes, with variable phenotypes and genotypes that affect the peripheral nervous system. These are rare diseases, which, in general, present progressive functional limitation secondary to muscle weakness, and may lead to the inability to sit, walk, perform manual activities, speak, swallow, and breathe, besides systemic complications secondary to possible damage to the heart muscle, bronchoaspiration, and respiratory infection, for example, besides musculoskeletal deformities, and death [1,2,3].

The care of patients with disabilities must be based on the ethical principles of beneficence, non-maleficence, autonomy, and justice, from the moment of communicating the diagnosis to end-of-life decision-making [4,5]. There is a gap in the literature related to breaking bad news regarding neurological diseases, each with different communication needs [6,7,8,9]. Feelings of stress during the conveyance of these diagnoses are described by physicians who highlight the need for training in communication techniques [6,7,10,11,12].

Improving communication skills strengthens the physician–patient/family bond, motivation and adherence to treatment, the feeling of security and hope, in addition to favoring shared decision-making, avoiding inappropriate expectations [7,13,14,15], optimizing consultation time [16], and available resources [15]. Poor communication has short- and long-term effects on the psychological and social health of patients and their families [15,17].

Given the above, the existence of a guide to assist in the communication of the diagnosis of neuromuscular diseases is essential to reduce stress and communication barriers between students, physicians in training, and experts, favoring the ethics of individual/family-centered care.

## 2. Methodology

A systematic review was performed by searching for articles on the Pubmed, Bireme, and Scopus websites until May 2022. The descriptors used in Pubmed were ((Muscular Disorders) OR (Neuromuscular Disease)) AND ((Bad news communication) OR (Truth Disclosure) OR (Breaking bad news)), in Bireme and Scopus were (“Muscular Diseases” OR “Neuromuscular Diseases”) AND (“Truth Disclosure” OR “Bad news communication” OR “Breaking bad News”). There was no restriction regarding the year of publication, and the filters used were studies in humans and English. The flowchart displaying the article selection is shown in Figure 1 [18]. Table 1 summarizes the inclusion and exclusion criteria.

Initially, a pre-selection of articles was conducted by the first author, followed by a final selection after analysis and discussion with the other authors, considering the interdisciplinarity inherent to the theme. After the stages of initial selection and final approval, the analysis of the articles began, independently reviewed, avoiding any ambiguity. Articles were not excluded based on the quality of the research presented, because of the small number of studies selected, and there was no superiority or need to exclude the experiences described.

The articles were carefully analyzed and individually interpreted through the following questions: 1. What factors have positively impacted the communication of the diagnosis of a neuromuscular disease? 2. Which communication barriers were described during the communication of this diagnosis? Except for some cases of basic numerical analysis of percentages, the data used for our research questions were qualitatively analyzed, after a synthesis of correlated results, and grouped through narrative textual synthesis, constituting a communication guide [19]. Results on the perspectives of patients, caregivers, and physicians were analyzed together due to the limited number of selected studies.

This guide was prepared after a neurologist with expertise in neuromuscular diseases, a specialist in medical law and bioethics, and a researcher specialized in palliative care and bioethics analyzed the results. Suggestions and critical remarks were sent by email to the first author, who conducted the final review and evaluation.

The following communication protocols for breaking bad news were used as a basis: SPIKES, the Six-Step Protocol for Delivering Bad News [20], EFNS guidelines on the Clinical Management of Amyotrophic Lateral Sclerosis (MALS)—revised report of an EFNS task force [16], National Institute for Health and Care Excellence (NICE) for the assessment and management of motor neuron disease [21], and a consensus issued by six Italian scientific societies and four parent associations on the communication of the diagnosis of genetic diseases and malformation syndromes [15].

## 3. Results

Sixteen articles were included for the final analysis after the selection of the systematic review. Five (31.2%) of them describe the perspective of people diagnosed with Amyotrophic Lateral Sclerosis (ALS), two (12.5%) report the experiences of family caregivers of people diagnosed with ALS, two (12.5%) of family caregivers of people with Duchenne muscular dystrophy (DMD), one (6.25%) of family caregivers of people with spinal muscular atrophy (SMA), three (18.75%) are mixed studies involving affected people and their caregivers (12.5% of ALS and 6.25% of chromosomal genetic diseases), two (12.5%) involve neurologists, and one (6.25%) resident physicians. One article obtained during the search in the references approaches the perspective of communication of the diagnosis between family members and people with genetic diseases, linked to the X chromosome, which, despite not directly addressing neuromuscular diseases, was selected for the final analysis because of the similarity related to rarity, genetic origin, functional impairment, absence of curative treatment, and need for continued rehabilitation [22]. The article on communication in the SMA was approved for publication and is under review [23].

The selected studies describe experiences from the United States of America, Canada, Europe, Australia, and Brazil (Table 2). Samples from Central America, Africa, Asia, and other South American countries were not found in the websites searched. There are several studies related to ALS, a few related to DMD, and only one to SMA, with a gap in the discussion about the perspective of the process of communicating the diagnosis for people with other neuromuscular diseases and their families, despite describing well the resulting negative impact of the poor communication of the diagnoses of neurodegenerative diseases and the low satisfaction with the communication process [6,7,10,11,12,15,17,24]. The studies described are summarized in Table 2.

The analysis of the results gave rise to the EMPATIA Communication Guide, subdivided into seven steps to support the transmission of the diagnosis of neuromuscular diseases, they are Empathy, Message, Prognosis, Acknowledgment, Time, Individualization, and Autonomy (Table 3).

## 4. Discussion

### 4.1. Step 1: E—Empathy

Empathy is the essence of this communication guide. Despite being considered a basic skill [5], several studies have described the absence of empathy during the diagnosis of neurodegenerative diseases [6,11,14,23,25,26,27,28]. Communicating the diagnosis of a neuromuscular disease to a patient and/or their family members requires empathy for the others and understanding on what, where, how, and in the presence of whom this should happen.

There must be medical preparation before the consultation so that breaking bad news does not result in a communication barrier [20,25,29] due to an uncertain prognosis, the absence of curative treatment, the anguish of not having all the answers, the fear of destroying hope, faith, or generate suffering, in addition to the reaction of patients and/or family members, and even the feeling of technical inability to transmit this diagnosis [11,20,32]. Given this, emotional involvement occurs in a bidirectional flow between physicians and patients/caregivers, since it is not a mere transfer of information [13].

Professionals must keep in mind that empathically communicating the truth does not harm the quality of life of their patients, and may also improve their vitality [22,33], reduce feelings of guilt and false beliefs, besides being an opportunity to discuss preferences, plan goals with shared decision-making, and promote a sense of control and security, empowering patients [5,6,32]. In contrast, inappropriate conversations may cause a more negative impact than all the struggle and hardship related to continued care [17].

The diagnosis must be informed in person, whenever possible [15,16,17,20], avoiding physical or electronic correspondence, telephone contact, or calls. This communication is a medical act (Art.34 of the Code of Medical Ethics) and therefore cannot be attributed to another health professional [34]. In the case of neuromuscular diseases, the neurologist, who has technical expertise on the subject, is the most suitable professional to transmit this information [21]. Avoiding weekends, nights, days before holidays, and commemorative dates, following the “rules of good practice” of breaking bad news is also recommended [4,7,8]. Patients must be instructed to invite a close companion or, if they are children, both parents or two family members is preferable [17,20,31], regarding consultations. This facilitates emotional support and understanding of the information, besides assisting in their continued care, which is often necessary [7,13,14,16,25,28,29].

Attention must be given to the feeling of privacy regarding the physical environment, the number of professionals and students during the breaking of bad news, and the opportunity for patients to get dressed, if they have been recently examined [15,16,20,31]. Privacy is mentioned as a satisfaction factor during the communication of the diagnosis of motor neuron disease [6,25,26,27,28]. Physicians must be attentive to non-verbal aspects, emotional state, personality, values, intellectual characteristics, and psychosocial and existential concerns, becoming flexible, abandoning any kind of prejudice, and assuming a posture centered on the interlocutor [12,15,25,28]. Thus, there is ethical and therapeutic value in using the first-person plural (we), referring to patients and family members by name, and adapting to the “pace of the patient” and their “psychic time” the amount and speed of information being given, so they can assimilate what was heard [13,15,16,25].

As tables or desks may be perceived as a communication obstacle, professionals must sit close to the patients, preferably side by side and at the same level, and look at them with care, indicating interest and allowing active listening [15,16,20]. Touching their arms or holding their hands, despite being considered an empathic posture, is not always desired by patients during the communication process [9,20] and, therefore, individual limits must not be disrespected. Interruptions must be avoided. Telephone calls, discussing topics related to other patients, and the need to prepare or sign various documents generate a feeling of poorly planned care [15,16,25,29]. The patient should be informed of possible time restrictions or expected interruptions [20].

### 4.2. Step 2: M—Message

The diagnosis must not be omitted. The “conspiracy of silence” or “pious lie” is a way of “sparing” the patient; or the distortion of information by, for example, using generalized terms such as “diseases of the nervous system” and “damage of motor nerves” instead of the actual name of the disease [6,13,16,29,35], which may cause feelings of helplessness and anguish, disturbing the moment of coping with the condition [30,35]. A sincere “I don’t know” is preferable to an unsubstantiated answer, generating a false interpretation and negative repercussions in the life of patients and their families [15].

The message must be communicated clearly and up to date, with accessible vocabulary, an appropriate tone of voice, uses of examples, metaphors, but also simple drawings or images. Sentimentality, ambiguity, understatement, medical jargon, and impersonal expressions must be avoided [7,8,9,12,13,15,16,21,25,26,27,28]. Exercising caution regarding the “over description” of clinical aspects or the scientific deepening of the information provided is needed, as the initial impact of the diagnosis compromises understanding [14,15,29]. Because they are rare diseases, most of the time patients and their families are unaware of them or only know them superficially [6,31], so it is interesting to ask what they know about it [15,16,20,21], understanding their expectations [17,21,25,28,29].

Physicians must also listen to the patients and make sure that the message transmitted was understood and, if there are any doubts, show patience, interest, and availability for further clarification [9,12,16,21,24,25,28,30] and avoid the “doorstep syndrome”, which is when, after providing all the information, they realize that patients and their family members have selectively filtered it [4,15]. Divergent opinions must not be confronted directly; the best strategy is understanding the experience and point of view of those involved [15], in the same way that one should not force, but guide care options [29].

Written referrals to rehabilitation services and other medical specialties, such as geneticists for genetic counseling, pulmonologists, orthopedists, among others, are recommended. Interdisciplinarity is essential for having a positive impact on the communication process, therapeutic planning, and the establishment of clinical and rehabilitation goals, improving functional independence, preventing complications, in addition to functioning as a support network, avoiding inappropriate conduct, financial expenses, and waste of time [7,8,9,11,15,21,22,25,26,27,28,29,30,31]. Thus, contact with the rehabilitation team must occur within 4 weeks of diagnosis [21].

Delivering a summary or providing materials related to the subject addressed in the consultation (explanations about the disease, classifications, possible causes, expected symptoms and how to manage them, treatment options, the procedure in case of complications, and ongoing research) is essential; professionals must also provide information on support groups, associations, labor rights, assistance benefits for people with disabilities, and reliable research sources [6,7,9,12,15,16,21,24,26,27,29,30]. Return consultations must be scheduled soon, ideally within 2 to 4 weeks [14,16] after the prior encounter. The search for information without guidance generates a “feeling of abandonment” [6,24], negatively impacting the acceptance of the diagnosis [7,9,14,26,27,29,35]. Scheduling the first return consultation with an interval of 14 days symbolizes a “moment to catch one’s breath”, process the information, adjust misunderstandings, and discuss doubts that arose after the consultation [29].

Physicians must maintain proper communication at each stage of evolution. Every conversation has to involve individualized support in the short, medium, and long term [4]; therefore, multiple meetings are essential [11,21,30,31]. Reassessments should ideally occur with the same neurologist every 2–3 months, but they can vary according to the diseases and their stages [16]. It is also important to maintain interdisciplinary communication from diagnosis to the end of life [4,30].

Breaking news for patients in childhood age:

The needs of the doctor/child/legal guardian trinomial must be individualized, with guidance to parents and family members who must have their doubts clarified to feel prepared to talk about the diagnosis in the different stages of maturity of their children [10,22,23,31]. The absence of this support causes psychological stress and increased predisposition to post-traumatic stress disorder in family caregivers [23]. In addition, the psychological vulnerability of parents associated with poor understanding of the various aspects related to the disease may generate inadequate decision-making [36].

“Keeping the child away from information” should be avoided as, for example, a “family secret”, since this attitude generates tension in the family environment, in addition to upset, frustrated, and anxious children and adolescents, with a continuous feeling of mistrust. The principle of truth, besides being an ethical stance, positively impacts the communication of the diagnosis [10]. Discussing the diagnosis gradually over time, but before puberty, with simple, direct, and age-appropriate information, using photos, books, and supporting materials is recommended. The approach depends on the personality of the child, and the level of affection and communication style of the family [22]. The autonomy of children and adolescents in the decision-making related to their diagnosis must be respected, understanding that cognitive maturity does not always correspond to the chronological one, but with the ability to abstract, understand information, and make decisions about their lives, according to the “mature minor theory” [37,38,39].

Physicians must listen to children, encourage them to ask questions, and share their feelings [22]. Open, honest, and frank dialogue results in children who are emotionally and psychologically more resilient and empowered [10], as well as avoiding inappropriate expectations [15]. Their strengths must be pointed out, valuing their skills and competencies [15]. They must also know that there are many people with the same diagnosis, all with unique challenges and differences, and that parents, physicians, and interdisciplinary teams will support them in coping with every issue [22].

In this context, attention is needed towards “those forgotten to be informed”, such as siblings of patients who, anguished, isolated, and silent, endure a family history of genetic and neurodegenerative disease [4], carrying feelings of guilt, fear, resentment, jealousy, and envy. Thus, genetic counseling is essential to favor family cohesion, reproductive decision-making, and the management of the feelings of both patients and family members [10].

### 4.3. Step 3: P—Prognosis

Another relevant recommendation is for professionals not to anticipate prognosis or determine a lifespan. Prognostic information should be individualized and realistic, understanding information needs and clarity [5,9,16,30], avoiding theoretical or generalized data [15]. Using survival or functional scales according to each disease and the clinical follow-up helps to understand their evolution; however, any uncertainties related to progression and estimated survival should be highlighted, explaining the individual variability, and stating that there are patients in better or worse situations [5].

Honesty about progression and acknowledging the severity of the disease, maintaining “realistic hope” through communication about neuroprotective [16] and disease-modifying [4] drugs, when appropriate, is essential; as well as ongoing research, medications, functional improvement, reduction of complications, goal planning, and increase in quality of life and life expectancy related to rehabilitation [7,16,29]. “Unrealistic optimism” can generate disproportionate expectations related to therapeutic goals and their associated risks, so the use of the term “cure” rather than “modification of the natural history of the disease” should be avoided [4,15,25,28]. Emphasizing positive aspects favors acceptance of the diagnosis [15,17,22], such as the preservation of cognition, keeping intellectual capacity intact, the possibility of making decisions, and, depending on the disease, studying, having children and some occupation, for example.

Physicians must teach, guide, and clarify doubts continuously at each stage of evolution, so that patients feel prepared in case of complications and safe to make therapeutic and palliative decisions, reducing the feeling of fear and anxiety [11,16,21,25,28,36]. In this way, continuing education reduces therapeutic obstinacy and, consequently, a slow and painful death [15]. It must be clear that patients are the protagonists of their stories and decision-making will be shared, their opinions respected, their wants and needs valued, and that they are free to hear a second opinion, if they wish [16,26,27,29]. They must also have the option of not knowing their prognosis, which can be discussed with the family if the patient allows it. If the physician judges they cannot make decisions because of cognitive impairments, their life expectancy should be discussed with the family [5].

Being sensitive to spiritual needs demonstrating compassion are also essential traits for health professionals. Even in the face of a bad prognosis, these behaviors may be sources of comfort, peace, hope, and belonging. When the medical team does not feel able to talk about spirituality or has divergent thoughts from the patient, spiritual counseling may help to reflect on faith, beliefs, and values [5,40]. Studies are needed to assess the impact of spiritual issues on quality of life and the desire for death in this population [16].

### 4.4. Step 4: A—Acknowledge

Another essential indispensable step is acknowledging fear, insecurity, pain, anger, denial, anxiety, outburst, revolt, sadness, or silence [16]. After clarifying the diagnosis and prognosis, it is imperative to support and comfort, being aware of the human being behind the neurodegenerative disease. The family must also feel welcomed, since, in general, participatory families are the greatest allies of the treatment. By valuing, so as not to minimize concerns, desires, feelings, and needs, showing interest, and providing the appropriate solutions and guidelines, physicians strengthen the patient/family-centered care, in addition to the trust in the physician–patient-family relationship [8,15,17,24].

Professionals must take care of their emotional health and acknowledge it. Their emotional reaction is an important communication barrier [12,25,28]; therefore, they must have resources to reduce or avoid the internalization of suffering related to the fear of causing anguish, communicating an unfavorable prognosis, or to the absence of a curative treatment [6].

### 4.5. Step 5: T—Time

Planning the step by step of this communication guide to adjust the time and reduce situations that show that the time is limited, such as interruptions and staring at the clock, is essential [16,29]. The duration recommended for these meetings is 45–60 min [14,16,25,28]. Therefore, professionals must keep the focus on the main theme, without straying into secondary dialogues, avoiding themes they cannot explain. Patients who had more time during the consultation reported greater satisfaction with the possibility of better digesting the information, conveying their feelings, and clearing up doubts, compared to those who had short consultations, which generated frustration and a feeling of haste [6,9]. Unavailability of time is recurrently described in the literature related to neurodegenerative diseases as an important communication barrier [6,7,9,11,12,14,20,21,25,26,27,28,30].

### 4.6. Step 6: I—Individualize

Physicians must also individualize the process, adapting this guide to the context related to the emotional, social, cultural, intellectual, financial, and religious issues of those involved and the information needs, centered on the patient and their family caregivers [8,25,28]. Each communication process is unique, and the physician must not have fixed protocols in mind but must be open to see if the initial information was sufficiently understood and expectations reached [15,20,31].

Judgment must be avoided and be replaced by guidance and support throughout the course of the disease [4,22]. Not all families will be able to conduct rehabilitation with a complete interdisciplinary team, whether for financial reasons, the location of health services, or the time required to provide care, but they must be well informed about the benefits of an effective treatment, as well as being encouraged, but not forced, to perform it [15,21,29]. If face-to-face follow-up is not possible, teleservice may be an option to reduce “therapeutic abandonment” and assist in support [41].

Adapting the service environment is also recommended, organizing tables and chairs, accessibility, noise reduction, and privacy, in addition to the training and guidance of a care team that involves all the staff, from the doorman to professionals who perform general services and secretaries, to facilitate accessibility and have empathic postures and conducts that value equity. The involvement of other health professionals in the integrated and continuous treatment is essential, besides the administration of the appointment schedule according to Step 2.8, and 2.9.

### 4.7. Step 7: A—Autonomy

The desired consequence of the EMPATIA guide is respect for the autonomy of those involved. Efficient communication favors consent and conscious and voluntary decision-making by patients or their legal guardians [42]. Professionals must make sure that the communication process results in autonomous individuals who, understanding the impact of their decisions, are free to decide according to their beliefs, perspectives, and values, even when opposing the opinion of their physicians [43].

## 5. Conclusions

This systematic review has demonstrated that empathic communication of the diagnosis and prognosis of a neuromuscular disease requires availability and management of time for individualized technical clarification according to the particularities of the patients and their caregivers. After the necessary information is given, there must be acknowledgment and guidance on which paths to follow in order to respect the autonomy of those involved.

In this way, the EMPATIA guide reflects the bioethical analysis of the literature related to breaking bad news about neuromuscular diseases, based on internationally supported protocols and a transparent and objective systematic review, discussed in an interdisciplinary manner. This guide fills the gap related to the communication of the diagnosis and the need for technical training of neurologists [6,7,8,12,14,26,27,29], favoring a satisfactory communication process [15,17].

## 6. Future Perspectives

The authors intend to analyze the impact of EMPATIA on several aspects, such as the communication of the diagnosis of neuromuscular diseases, the physician–patient relationship, the stress related to the doctor during this moment [6,12,26,27], the psychological sequelae and post-traumatic stress disorder in family members, and, consequently, on the autonomy of those involved during the decision-making [23].

## 7. Limitations

The limitations of this guide comprise the lack of research related to the communication process of the diagnosis for most neuromuscular diseases and the absence of this type of investigation in several continents, which may require cross-cultural adaptations or specific adjustments regarding some neuromuscular diseases.

## Figures and Tables

**Figure 1 ijerph-19-09792-f001:**
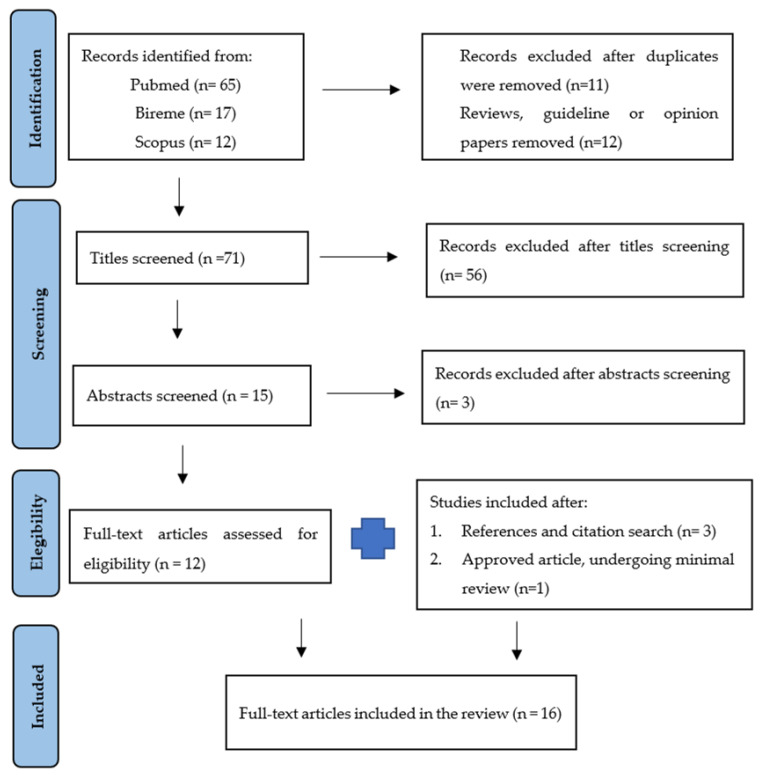
PRISMA Diagram.

**Table 1 ijerph-19-09792-t001:** Inclusion and Exclusion Criteria.

Inclusion Criteria	Exclusion Criteria
Original research articles Written in English Peer-reviewed publications Monogenic neuromuscular disease	Case reports, reviews, and opinion papers Written in another language Publications in unreviewed journals Rapidly reversible neuromuscular disorders, such as transient neuropathies and myopathies
Studies related to the moment of communicating the diagnosis of a neuromuscular disease, addressing factors related to the disclosure of this diagnosis, communication barriers, and disclosure of the prognosis.	Studies describing screening tests and therapeutic decisions.

**Table 2 ijerph-19-09792-t002:** Summary of studies selected in the systematic review.

N.	Study	Study Proposal	Design	Methodology	Sample	Factors that Positively Impact the Communication of the Diagnosis of Neuromuscular Diseases	Communication Barriers in the Diagnosis of Neuromuscular Diseases	Study Conclusions
1	Fernandes, 2022 [23]	Analyzing the principle of autonomy in the communication of the diagnosis of spinal muscular atrophy (SMA)	Quantitative	Online questionnaire	50 family caregivers from AME, Brazil.	Empathic relationship between doctor, child, and family.	Lack of guidance to parents on how to communicate the diagnosis to their children	The communication of the SMA diagnosis to the child or adolescent was predominantly carried out by parents who did not feel prepared to discuss the topic, resulting in psychological sequelae and increased predisposition to post-traumatic stress syndrome.
2	Anestis 2021 [7]	Assessing current practice and perspectives from British neurologists on diagnosing a motor neurodegenerative disease	Qualitative and quantitative	Online questionnaire with demographic information, current practice, bad news experience, and training needs	49 British Neurologists	Be honest without giving up hope Family participation Involvement of a multidisciplinary team	Fear of causing distress Young patient Reduced consultation time Big time gap in scheduling the return consultation Absence of curative treatment or disease-modifying drug Excessive workload	Although compliance with basic standards of good practice was reported by most professionals, the study identified several areas for improvement, in addition to the need for training of professionals regarding the breaking of bad news.
3	Mirza, 2019 [9]	Determining whether guidelines for breaking bad news, particularly SPIKES, are consistent with the preferences of patients diagnosed with cancer, lupus, amyotrophic lateral sclerosis, multiple sclerosis, HIV/AIDS, or Parkinson’s disease.	Qualitative and quantitative	Online questionnaire	1337 patients from Canada	Empathetic doctors, no rush Convey a feeling of availability Clear explanation of diagnosis, prognosis, and treatment Make sure the patient understands the information Written information Contact support organizations, groups, or advisors Small time gap for the return consultation	Some patients do not want to be touched by doctors	It highlights that SPIKES reflects the perspectives of many patient groups and identifies five additional suggestions to help clinicians break bad news.
4	Aoun, 2016, 2018 [11,25] **	Describing experiences of receiving a diagnosis of motor neuron disease (MND) in order to take a more person-centered approach	Qualitative and quantitative	Anonymous postal questionnaire sent via patients association	248 people diagnosed with NMD from Australia	Honest, direct, receptive, caring, kind, compassionate, empathetic, concerned, practical, and engaged neurologist. Empathic response to patient/family feelings Clear information and suggestion of realistic goals Understand patient/family expectations Continuing Guidance Time availability (45–55 min) Privacy Psychological support Contact with associations Multidisciplinary team	Absence of a family member at the consultation Excessive interruptions during the consultation Absence of opportunity for discussion Lack of emotional support Emotional reaction of the neurologist “Prosaic”, “very clinical”, “isolated”, and “insensitive” neurologist	The person-centered approach, focused on values, preferences, psychosocial and existential concerns, caring, supporting, and comforting rather than just treating is what matters most in a relatively short period of a fatal illness.
5	O’Connor, 2018 [26]/Aoun 2017 [27] **	Describing the experiences of family caregivers of people with NMD when receiving the diagnosis, to improve the communication of the diagnosis	Qualitative	Anonymous postal questionnaire sent via patients association	190 Australian family caregivers	Empathetic neurologist who responds appropriately to your emotions Technical knowledge Long queries Privacy Written information Referral for a second opinion Referral to patient associations Sensitive, warm, and respectful neurologist	Short consultation time Big time gap for scheduling the return consultation Lack of clarity regarding evolution and prognosis Lack of empathy and blunt delivery took away the sense of hope	Neurologists can benefit from education and training in communication skills to respond to emotions of patients and families and develop best practice protocols.
6	Bendixen, 2017 [8]	Reflections from parents on the Duchenne muscular dystrophy (DMD) diagnostic process	Qualitative	Semi-structured interview conducted by telephone	15 parents of boys withDMD from the USA	Interdisciplinary team support	Limited medical knowledge Minimized or disregarded parental concerns Tactless communication	Despite marked medical progress in recent decades, there are still barriers to diagnosing DMD and guiding care.
7	Aoun, 2016 [28]	Determining the practice in communicating the diagnosis of NMD, training needs in breaking bad news, and comparing neurologists’ experiences with patients seen in the same year and with international protocols	Qualitative and quantitative	Online questionnaire	73 neurologists from Australia	Availability of time Empathic responses Multidisciplinary approach Continued support Contact with associations	Fear of causing distress Lack of effective treatment Fear of not having all the answers Lack of training or need for improvement in breaking bad news	Neurologists reported some difficulty in transmitting the diagnosis of NMD, with a feeling of moderate to intense stress, and demonstrated interest in training to respond to the emotional demand of patients and to develop a protocol of good practices.
8	Seeber, 2016 [29]	Evaluate how NMD patients react to the disclosure of the diagnosis in 2 consultations with an interval of 14 days	Qualitative	Semi-structured household interview and report analysis	21 patients from the Netherlands	Confirmation of the diagnosis and communication of the truth after a period of uncertainty Early return consultation with the same doctor symbolized “time catch one’s breath” Early feedback facilitated adjustment of misunderstandings and discussion of involvement of other bodies Having your desires and expectations valued Contact with the specialized team Support from family and friends Option to take part in clinical research Listen to the opinion of the neurologist about ongoing research Offering but not forcing care options	Impact of the diagnosis limits the questioning initially Difficulty of the neurologist in transmitting the diagnosis	This approach fits well with the way patients process the message, gather information, and deal with their altered perspective on life. The neurologist must follow a clear protocol and schedule a short-term return consultation.
9	Dennis, 2015 [22]	Assessing the timing and content revealed to a child by their parents regarding the diagnosis of aneuploidies, the resources accessed, feelings and concerns of the parents, and recommendations for approaches of breaking bad news	Qualitative and quantitative	Online questionnaire	139 relatives and 67 people with sex chromosome aneuploidies from the USA, Canada, Europe, and Australia	Discuss the diagnosis gradually over time before puberty Well-informed parents to answer their own questions Be honest, open, and calm Simple, direct, and age-appropriate information Use of photos, books, and support materials Be positive and non-judgmental Everyone has unique challenges and differences Point out the strengths of the children Encourage the child to ask questions and discuss their feelings Inform that there are many people with the same diagnosis Emphasize parental support in coping with problems Support from a health professional in the moment of breaking bad news	More comfort in discussing with the physician than with the parents Lack of medical support for parents in the process Fear of the child feeling different and affecting their self-esteem	The approach depends on the personality, family affection level, and communication style of the children. Their reactions were commonly neutral or positive. Negative reactions may have resulted from inappropriate information considering their maturity.
10	Schellenberg, 2014 [12]	Examine the ability of resident physicians in reporting ALS diagnosis	Quantitative	Checklist on breaking bad news	22 Canadian resident physicians		Need for training in communication techniques Inadequate vocabulary Inappropriate non-verbal communication Reduced time Difficulty summarizing information, asking questions, and confirming understanding Medical inability to deal with their own emotions	Dissatisfaction and apprehension of resident doctors regarding the training for communicating this diagnosis
11	Chió, 2008 [24]	Assessing information preferences and information-seeking behavior in ALS patients and caregivers	Quantitative	Interview using a structured questionnaire	60 patients and 20 caregivers from Italy	Reporting on current research, disease-modifying therapies, and ALS outcomes Understanding what patients and caregivers want to know and encouraging them to ask accurate questions Feeling that the doctor understood the feelings of the patients	Low satisfaction and high stress during the communication process resulted in an alternative search for information	Healthcare professionals should be aware that ALS patients and caregivers often use the internet for information and should help them select and interpret the news they have encountered.
12	McCluskey, 2004 [14]	Determining the degree of satisfaction of patients and/or family members with the way they received the diagnosis of ALS and to assess aspects associated with greater satisfaction	Quantitative	Anonymous postal questionnaire sent via patient association	163 ALS patients and/or family members fromPhiladelphia, USA	Discussing end-of-life behaviors at first had no negative impact Minimum of 45 min of consultation Presence of family members Early return consultation	Detailed and specific information is not well understood at first due to the shock of the diagnosis	There is the possibility of improving the communication of the diagnosis of ALS, with adherence to the techniques of breaking of bad news and availability of time during the consultation.
13	Beisecker, 1988 [30]	Perspectives of the ALS patients on the role of physicians and other health professionalsin their care	Qualitative	Semi structured interview	41 ALS patients from Kansas, USA	Direct, honest, but not premature communication Physician who is sensitive to the readiness of patients and who conveys hope Provision of written materials	Lack of opportunity to pose questions Misinformation or withholding information Inadequate disclosure style Disagreement about the treatment plan Lack of time	Patients want to see the doctor at every appointment and expect emotional support and care from both the doctor and the nurse and related healthcare professionals.
14	Firth, 1983 [31]	Sharing the experiences of parents of children with DMD at the time of diagnosis	Qualitative	Home interview	66 families (single mother or father and mother) from Great Britain	Presence of both parents Early return consultation Possibility of several meetings Providing contact with specialized support team Providing contact with families with similar experience	Delay in diagnosis Number of students in the environment	There is not just one way to tell parents that their children have DMD. The article summarizes suggestions from parents who have had this experience.

** Articles with similar methodology and results.

**Table 3 ijerph-19-09792-t003:** EMPATIA Communication Guide.

Communication Guide for the Diagnosis of Neuromuscular Diseases	
Empathy	Be empathetic Prepare yourself technically and emotionally Convey the diagnosis in person Avoid weekends, nights, days before holidays and commemorative dates Encourage the presence of family members Respect the privacy of the moment Pay attention to verbal and non-verbal expressions Sit close to the patient, keeping your gaze direct and continuous Avoid interruptions
Message	Do not omit or distort the diagnosis Convey information in a clear, honest, up-to-date, accessible, and unambiguous manner, without euphemism, jargon, sentimentalism, and with an appropriate tone of voice Avoid being too technical at first Listen to the patient and understand their expectations, making yourself available Do not directly confront differing opinions Refer the rehabilitation team and specialized services Summarize key information and provide materials with diverse guidance related to the illness Schedule a return consultation between 2–4 weeks to also approach questions written down by patients/family members. Maintain proper communication at each stage of evolution Particularities for patients in childhood age: Individualize the needs of the physician/child/legal guardian trinomial Guide and support parents in avoiding “information deprivation”, as treating the diagnosis and information connected to it as a “family secret” Adapt communication to the level of maturity of the children Encourage the children to ask questions and talk about their feelings Point out the strengths of the children, inform that there are others with the same diagnosis, and emphasize the support of the parents for dealing with problems Beware of “Forgotten Information Individuals”
Prognosis	Do not anticipate the prognosis or determine a lifespan Keep “realistic hope” and not “unrealistic optimism” Emphasize positive aspects Conduct a continuous education regarding procedures and possible complications Value wants and needs of every patient Be sensitive to spiritual needs and show compassion Respect the right not to know, if applicable, and ask for the name of someone who can share treatment decisions with the physician
Acknowledgment	Acknowledge fear, pain, anger, denial, anxiety, outbursts, crying, and silence Value feelings and doubts, without minimizing them, but looking for solutions and answers Acknowledge suffering from others without internalizing it
Time	Keep a communication plan in mind Avoid discussing topics that have no connection with the diagnosis or that you cannot explain The recommended time for this meeting is between 45–60 min
Individualization	Adapt to the social, cultural, intellectual, and religious particularities of the patient and their companions Avoid judgments Adapt the environment where the meeting is held
Autonomy	Respect the autonomy of those involved Ensure that the information and clarification provided resulted in free and informed decisions

## Data Availability

Not applicable.

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
