# Peer review of "EMPATIA: A Guide for Communicating the Diagnosis of Neuromuscular Diseases"

_ijerph, 2022, doi:10.3390/ijerph19169792_

Round 1
Reviewer 1 Report
The review is nicely written, and very informative as far as this topic is concerned. Nevertheless, there are some shortcomings, especially lacking statistical information.
I have a question. Line 111-112. "The selected studies describe experiences from the United States of America, Canada, 111 Europe, Australia, and Brazil. They do not include samples from Central America, Africa, 112 Asia, and other South American countries."
Why did the authors omit some countries? Is it a cultural reason? Is a lack of similar studies?
I think a comment explaining why those countries were excluded should be added to the manuscript.
Author Response
Response to Reviewer 1 Comments
Point 1: The review is nicely written, and very informative as far as this topic is concerned.
Response 1: I am very honored and happy.
Point 2: Nevertheless, there are some shortcomings, especially lacking statistical information.
Response 2: “Except for some cases of basic numerical analysis of percentages, the data used for our research questions were qualitatively analyzed, after a synthesis of correlated results, and grouped through narrative textual synthesis, constituting a communication guide (19). Results on the perspectives of patients, caregivers and physicians were analyzed together, due to the limited number of selected studies.”
Point 3: I have a question. Line 111-112. "The selected studies describe experiences from the United States of America, Canada, 111 Europe, Australia, and Brazil. They do not include samples from Central America, Africa, 112 Asia, and other South American countries." Why did the authors omit some countries? Is it a cultural reason? Is a lack of similar studies? I think a comment explaining why those countries were excluded should be added to the manuscript.
Response 3: “Samples from Central America, Africa, Asia, and other South American countries were not found in the websites searched”
Additional review:
- When submitting the article, there was an error in the formatting of the Prisma diagram. I made the adjustment.
- Added table 2 outside the main text. Now I included the table in the same file.
Reviewer 2 Report
The article is well written, and it covers a space that was deserted.
I recommend that the authors synthesize the discussion a little more to make it more understandable, and increase the conclusion to make clear the information that they want to convey.
Thank you!
Author Response
Response to Reviewer 2 Comments
Point 1: The article is well written, and it covers a space that was deserted. I recommend that the authors synthesize the discussion a little more to make it more understandable,
Response 1: I am very honored and happy with your comment. I tried to reorganize the discussion. If you believe there is any further improvement, I am available to improve it. Thanks.
Point 2: increase the conclusion to make clear the information that they want to convey. Thank you!
Response 2: “This systematic review has demonstrated that empathic communication of the diagnosis and prognosis of a neuromuscular disease requires availability and management of time for individualized technical clarification according to the particularities of the patients and their caregivers. After the necessary information is given, there must be acknowledgment and guidance on which paths to follow to respect the autonomy of those involved.” ...
Additional review:
- When submitting the article, there was an error in the formatting of the Prisma diagram. I made the adjustment.
- Added table 2 outside the main text. Now I included the table in the same file.
Reviewer 3 Report
Thank you for the possibility to review the manuscript entitled: "EMPATIA: A guide for communicating the diagnosis of neuro-2 muscular diseases".
This review aimed to develop a guide to reduce communication barriers in the diagnosis of neuromuscular diseases.
However, some revisions are required before its acceptance:
Revise all the manuscript in English language
Improve the discussion section with more references and in a more discursive and harmonious approach.
Author Response
Response to Reviewer 3 Comments
Thank you for the possibility to review the manuscript entitled: "EMPATIA: A guide for communicating the diagnosis of neuro-2 muscular diseases". This review aimed to develop a guide to reduce communication barriers in the diagnosis of neuromuscular diseases. However, some revisions are required before its acceptance:
Point 1: Revise all the manuscript in English language
Response 1: The article was proofread by an official translator.
Point 2: Improve the discussion section with more references and in a more discursive and harmonious approach.
Response 2: I tried to reorganize the discussion. If you believe there is any further improvement, I am available to improve it.
Additional review:
- When submitting the article, there was an error in the formatting of the Prisma diagram. I made the adjustment.
- Added table 2 outside the main text. Now I included the table in the same file.